# OpenReview forum: "MPSelectTune: Prompt-type Selection for Fine-tuning improves Concept Unlearning in LLMs"
_NeurIPS.cc/2025/Workshop/Reliable_ML — NeurIPS 2025 - Reliable ML Workshop_

### Official Review · Reviewer_dHpq · 2025-09-10
**A Promising Framework for Robust Concept Unlearning via Prompt-Type Selection**

**Rating:** 8
**Confidence:** 4

**Review:**

This paper introduces MPSelectTune, a two-stage fine-tuning framework for concept unlearning in large language models (LLMs). The work addresses an important gap: existing unlearning methods often neglect the impact of prompt type, even though concept leakage in LLMs is highly prompt-dependent. The proposed approach first performs multi-prompt fine-tuning (MPTune) with a multi-task loss, then fine-tunes again using the worst-performing prompt type to adversarially minimize concept retention.

Strengths
* Novel perspective on prompt types: The idea of explicitly leveraging the worst-case prompt type is simple yet powerful, and well-justified both theoretically and empirically.
* Comprehensive evaluation: The authors benchmark across diverse datasets (Bios, RT-Gender, ToxicBias, Adult Census, and SciQ-WMDP-Bio) and multiple model families (LLaMA-2, LLaMA-3.1, Mistral). This breadth adds strong credibility.
* Clear improvements over baselines: Results show consistent reductions in concept accuracy (up to −17%) without sacrificing main task performance. The spuriousness-score analysis is particularly compelling, showing large reductions (23–74%).
* Generalization to unseen prompts: The study includes evaluations on prompt types not seen during training, a critical but often overlooked dimension of robustness.

Weaknesses / Concerns
* Prompt selection automation: The method requires identifying the “worst prompt type” via evaluation on training data, which may not scale or adapt dynamically. Future work could consider automated or online prompt-type selection strategies.
* Limited scope of concepts: Experiments focus primarily on binary concepts (e.g., gender, race, bio-weapon knowledge). It remains unclear whether the method extends as effectively to multi-class or more abstract concept unlearning.
* Complexity of training: The two-stage fine-tuning process, while effective, doubles the training overhead compared to single-stage approaches. A discussion on efficiency trade-offs would be valuable.
* Interpretability of unlearning: While reductions in accuracy are reported, there is less discussion on why certain prompt types are consistently “worst” predictors. Additional analysis could help connect prompt construction strategies to unlearning difficulty.

---

### Official Review · Reviewer_qtC8 · 2025-09-17
**Peer review of MPSelectTune**

**Rating:** 7
**Confidence:** 4

**Review:**

Summary:
The authors propose a new algorithm, MPSelectTune, which is a fine-tuning algorithm for concept unlearning tasks to remove biased and/or harmful concepts. The algorithm works in two stages, multi-prompt and select tuning, which constrains the model with a novel loss function then performing adversarial learning on the worst prompt type respectively. The results show a higher performance in terms of tasks, but near random performance on concepts which shows a strong movement towards unlearning the concepts.

Strengths:
The ethical significance of the problem addressed is noteworthy. Successful concept unlearning could permit the use of imperfect data used to train these models while enabling the model to give more reliable answers. The paper provides a novel approach for concept unlearning that offers a coherent framework that produced promising results also including unseen data. The method is well-defined and presented clearly. Results present comparisons to baselines to show the significant improvement over current methods. The authors also provided numerous examples to give insight into what the results actually represent.

Weaknesses:
There isn't much discussion about the extra compute required by the two stage process of the algorithm. It is shown in Table 10 that to run both stages may nearly double the runtime relative to the compared methods. Additionally, the major limitation mentioned concerning the selection of prompt types is significant as the algorithm depends on finding the appropriate prompts to minimize accuracy. If the necessary worst-case examples are not present, it could affect performance in terms of concept unlearning.

Suggestions:
Mentioning the increased compute time would be helpful in the limitations. This is an area of possible improvement since there is some tradeoff of time and effectiveness for using this method compared to the baselines mentioned. Another addition that could improve the study is observing where the method is likely to fail (e.g. with a poor selection of prompts) since the authors mention how prompts are selected is a limitation of the study. This could spark interest into what future research may entail.